# Evaluation of Chitosan Derivatives Modified Mesoporous Silica Nanoparticles as Delivery Carrier

**DOI:** 10.3390/molecules26092490

**Published:** 2021-04-24

**Authors:** Qi Li, Wenqian Wang, Gaowei Hu, Xianlan Cui, Dejun Sun, Zheng Jin, Kai Zhao

**Affiliations:** 1Key Laboratory of Chemical Engineering Process and Technology for High-efficiency Conversion, College of Chemistry and Material Sciences, Heilongjiang University, Harbin 150080, China; liqi114013@163.com (Q.L.); wangwenqian3104@163.com (W.W.); 2Institute of Nanobiomaterials and Immunology, School of Life Science, Taizhou University, Taizhou 318000, China; hugaowei68@163.com; 3Key Laboratory of Microbiology, School of Life Science, College of Heilongjiang Province, Heilongjiang University, Harbin 150080, China; xianlan_cui@163.com; 4Engineering Research Center of Agricultural Microbiology Technology, Ministry of Education, Heilongjiang University, Harbin 150080, China; 5Bluesky Biotech (Harbin) Co., Ltd., Harbin 150028, China; 6Heilongjiang Kaizhenglihua Biological and Chemical Technology Co., Ltd., Harbin 150080, China

**Keywords:** mesoporous SiO_2_ nanoparticles, *N*-2-Hydroxypropyl trimethyl ammonium chloride chitosan, carboxymethyl chitosan, nanoparticle, delivery system

## Abstract

Chitosan is a non-toxic biological material, but chitosan is insoluble in water, which hinders the development and utilization of chitosan. Chitosan derivatives *N*-2-Hydroxypropyl trimethyl ammonium chloride (*N*-2-HACC) and carboxymethyl chitosan (CMCS) with good water solubility were synthesized by our laboratory. In this study, we synthesized mesoporous SiO_2_ nanoparticles by the emulsion, and then the mesoporous SiO_2_ nanoparticles were modified with γ-aminopropyltriethoxysilane to synthesize aminated mesoporous SiO_2_ nanoparticles; CMCS and *N*-2-HACC was used to cross-link the aminated mesoporous SiO_2_ nanoparticles to construct SiO_2_@CMCS-*N*-2-HACC nanoparticles. Because the aminated mesoporous SiO_2_ nanoparticles with positively charged can react with the mucous membranes, the virus enters the body mainly through mucous membranes, so Newcastle disease virus (NDV) was selected as the model drug to evaluate the performance of the SiO_2_@CMCS-*N*-2-HACC nanoparticles. We prepared the SiO_2_@CMCS-*N*-2-HACC nanoparticles loaded with inactivated NDV (NDV/SiO_2_@CMCS-*N*-2-HACC). The SiO_2_@CMCS-*N*-2-HACC nanoparticles as delivery carrier had high loading capacity, low cytotoxicity, good acid resistance and bile resistance and enteric solubility, and the structure of NDV protein encapsulated in the nano vaccine was not destroyed. In addition, the SiO_2_@CMCS-*N*-2-HACC nanoparticles could sustain slowly released NDV. Therefore, the SiO_2_@CMCS-*N*-2-HACC nanoparticles have the potential to be served as delivery vehicle for vaccine and/or drug.

## 1. Introduction

Drug delivery agents can regulate the distribution of drug in the body in terms of delivery space, time and dosage. The goal is to deliver the right amount of medicine to the right place at the right time to increase the bioavailability of the medicine and reduce costs and side effects [1,2,3].

The emergence of mesoporous structural materials has attracted the attention and research of scientists [4]. Mesoporous SiO_2_ nanoparticles have been proved to be the most promising and potential materials in the field of biomedicine by virtue of their unique porous structure and excellent biocompatibility [5,6]. Mesoporous SiO_2_ has a typical honeycomb channel structure and significant loading and retention capabilities. The size of SiO_2_ nanoparticles is generally between 250–400 nm [7]. It is a very good drug delivery system, especially as a nanocarrier for drugs, gene delivery, and bioimaging and photodynamic therapy [8,9,10]. However, SiO_2_ nanoparticles still have shortcomings. For example, it is still difficult to prepare mesoporous SiO_2_ nanoparticles with uniform particle size, monodisperse, and controllable size. The safety of mesoporous SiO_2_ has also become one of the main obstacles in clinical medicine [11,12]. The surface of the mesoporous SiO_2_ nanoparticles contains a large number of hydroxyl groups, and the surface energy is high, which is prone to agglomeration and not easy to disperse. Therefore, in order to reduce its surface energy and increase its dispersion rate, mesoporous SiO_2_ nanoparticles can be modified [13,14,15,16,17,18,19,20]. The modification methods of mesoporous SiO_2_ nanoparticles can be divided into two types: physical modification and chemical modification. Physical modification mainly uses ultraviolet and plasma to modify mesoporous SiO_2_ nanoparticles through physical effects such as adsorption and wrapping. The chemical modification is mainly through the reaction with the coupling agent, and the functional groups are grafted onto the mesoporous SiO_2_ nanoparticles by covalent bonding. Commonly used modification methods are the co-condensation method and post-grafting method in the chemical modification method. After surface functionalization of mesoporous SiO_2_ nanoparticles, their effectiveness as a drug delivery agent is also improved [21,22,23,24,25,26]. Mesoporous SiO_2_ nanoparticles were modified by 3-aminopropyltriethoxysilane, compared with unmodified mesoporous SiO_2_ nanoparticles, the surface-modified mesoporous SiO_2_ nanoparticles have better performance. The sustained-release effect is obvious, and it has better biocompatibility and bioavailability [27]. Surface-functionalized mesoporous SiO_2_ nanoparticles can control drug release, maintain a constant release behavior for a period of time, and prolong the number of repeated administrations of a fixed dose and a fixed time [28,29,30,31,32]. SiO_2_ is widely used in biomedicine. Large pores of mesoporous silica nanoparticles allow the other particles to be filled with a drug through endocytosis. These emerge as a highly attractive class of drug and gene delivery vector [33]. Li et al. used mesoporous silica nanoparticles as a delivery carrier, co-loaded anti-angiogenesis agents and chemotherapeutic drugs, and combined with targeting molecules for anti-angiogenesis and chemotherapy, the results showed that the dual-load drug delivery system could deliver the drugs in tumor blood vessels and entered into the tumor through a differentiated drug release strategy, thereby enhancing the anti-tumor effect [34]. However, mesoporous SiO_2_ nanoparticles alone have no mucosal adhesion, so chitosan derivative is used to modify SiO_2_ nanoparticles in this study.

Chitosan derivatives show good adhesion and permeability and can improve drug delivery and sustained release. Chitosan nanoparticles have important applications in targeting, slow-release and increased drug absorption. At present, chitosan derivative nanoparticles are mainly used for sustained release and targeted preparations of drugs, special preparations of proteins and polypeptides, and carriers for gene therapy [35].

Newcastle disease (ND) is an infectious disease of poultry. All tissues and organs of sick poultry infected with Newcastle disease virus (NDV) have the virus, and brain, spleen and lung of sick chickens have the highest toxic contents. NDV can enter the body from the digestive tract, respiratory tract, conjunctiva and trauma, and flow from the blood to the whole body within one day after infection, causing viremia. Therefore, the development of mucosal vaccination is one of the effective methods to prevent ND [36]. Mucosal vaccine can carry out antigen presentation through the gastrointestinal mucosa and stimulate gastrointestinal mucosal immunity. Compared with traditional injection vaccines, mucosal vaccines have better compliance, and the route of administration is simple and convenient. It can be administered orally or nasally. It is an ideal route of administration. However, its disadvantage is that the protein or adjuvant in the vaccine is easily decomposed in the gastrointestinal juice, thus reducing the immune effect. Therefore, it is necessary to develop a highly effective vaccine adjuvant and delivery system that can be taken orally to improve the efficacy of inactivated NDV vaccine. The application of mesoporous SiO_2_ nanoparticles in the field of delivery has been extensively explored [37,38,39]. The main reason is that mesoporous SiO_2_ nanoparticles has the advantages of uniform pore size, large surface area, large pore volume and strong drug-carrying capacity [34,40,41,42]. However, the tolerance of mesoporous SiO_2_ nanoparticles and aminated mesoporous SiO_2_ nanoparticles remains to be studied. In order to improve the tolerance of aminated mesoporous SiO_2_ nanoparticles, CMCS and *N*-2-HACC are used to double-modify it, while also protecting the SiO_2_ nanoparticles.

## 2. Materials and Methods

### 2.1. Materials

Chitosan (Degree of deacetylation 80–85%, the intrinsic viscosity was 37.9 L/g, and the average molecular weight of chitosan was 53.0 kDa. The main raw materials were shrimp shells and crab shells.), tetraethylorthosilicate, cetyltrimethylammonium bromide (≥99%), and formaldehyde were purchased from Shanghai Jingchun Biochemical Co., Ltd. Company in China. *N*-2-HACC and CMCS were synthesized by our group, the substitution degree of *N*-2-HACC used in the study was 60–65%, degree of deacetylation was 80–85%, the intrinsic viscosity is 15.4 L/g, molecular weight was 20.0 kDa; The substitution degree of CMCS was 90–95%, degree of deacetylation was 80–85%, the intrinsic viscosity was 2.4 L/g, molecular weight was 30.3 kDa [43]. Ammonia (25%) was purchased from Jinan Xinzhenyuan Chemical Company in China; γ-aminopropyltriethoxysilane (≥99.5%) was purchased from Hengdazhongcheng Technology Company in China; CCK-8 kit was purchased from Yisheng Biotechnology; pepsin and pancreatin were purchased from Ruji Biotechnology.

### 2.2. Synthesis of Mesoporous SiO_2_ Nanoparticles

Firstly, the water bath was heated to 60 °C and the speed set to 200 r/min. Then 120 mL of absolute ethanol, 100 mL of deionized water and 4.6 mL of 25% ammonium hydroxide were added into a 500 mL three-necked flask and stirred for 1 min; CTAB was added and stirred for 10 min; 1 mL of ethyl orthosilicate (TEOS) added and the reaction conditions kept for 12 h. After the heating was stopped, the reaction was kept for 12 h at room temperature. The reaction solution was poured into a high-temperature reaction kettle, and the reaction continued for 3 h in an oven at 180 °C. The reaction kettle was then placed at room temperature and cooled to room temperature. The product was collected using a vacuum filter and a filter membrane with a pore size of 0.45 μm was selected. The product was repeatedly washed with absolute ethanol three times and dried at 50 °C for 1 h. The dried product was placed into a muffle furnace, heated for 6 h at 700 °C, and then cooled to room temperature. The collected product was mesoporous SiO_2_ nanoparticles.

### 2.3. Synthesis of Aminated Mesoporous SiO_2_ Nanoparticles

A hundred milligrams of the mesoporous SiO_2_ nanoparticles were accurately weighed and added to 500 mL three-necked flask containing 100 mL of absolute ethanol and stirred for 1 min under the conditions of a water bath at 80 °C and 200 r/min and condensed and refluxed. A milliliter of γ-aminopropyltriethoxysilane was added to the above solution, and the reaction was maintained at 200 r/min in a water bath at 80 °C for 3 h. The filter paper with a diameter of 0.45 μm was selected for suction filtration, dried in an oven at 50 °C for 3 h, and the product was then collected. We used Fourier infrared spectroscopy to verify the amino (Spectrum RX-1, Perklin Elmer, Boston, USA).

### 2.4. Preparation of Aminated Mesoporous SiO_2_ Nanoparticles Loaded with Inactivated Newcastle Disease Virus

A certain weight of aminated mesoporous SiO_2_ nanoparticles (3 mg, 4 mg, 5 mg, 6 mg, and 7 mg) was taken and dissolved in 4 mL deionized water at room temperature and stirred mechanically at 1000 r/min for 5 min. A milliliter of NDV inactivation solution (the virus content is 7.504 mg/mL) was added dropwise, and mechanically stirred at 1000 r/min for 30 min at room temperature to obtain NDV/SiO_2_ nanoparticles. A kit was used to measure the encapsulation efficiency of NDV/SiO_2_ nanoparticles (Thermo Fisher Scientific, New York, NY, USA), and Zeta potential analyzer (BI-Zeta PALS, Brookheven Instruments, New York, NY, USA) was used to measure the potential of the nanoparticles to determine the ratio of aminated mesoporous SiO_2_ nanoparticles to the Newcastle disease virus inactivation solution. The encapsulation efficiency was calculated using the following equation:EE (%) = [(W_0_ − W_1_)/W_0_] × 100%(1)
LC (%) = [(W_0_ − W_1_)/W] × 100%(2)
where W_0_ is the total NDV mass added, W_1_ is the NDV mass in the supernatant, W is the total mass of drug-loaded nanoparticles.

### 2.5. Synthesis of NDV/SiO_2_@CMCS-N-2-HACC

A total of 5 mg of aminated mesoporous SiO_2_ nanoparticles were dissolved in 4 mL of deionized water, stirred for 5 min at 1000 r/min at room temperature. Then, 1 mL of NDV inactivation solution (virus content 7.504 mg/mL) was added dropwise into the above solution. Under the condition of stirring at 1000 r/min, 3 mL of 0.5 mg/mL CMCS solution was added dropwise to the above solution, and then 4 mL 0.5 mg/mL *N*-2-HACC solution was also added dropwise, with mechanical stirring for 10 min. The order of dropping was determined according to the results of Zeta potential. The aminated mesoporous SiO_2_ nanoparticles were positively charged, so the negatively charged CMCS solution was added dropwise first and the positively charged *N*-2-HACC solution was then dropped. It was stirred for another 20 min after the addition was complete. After the above reaction was completed, it was centrifuged for 30 min at 4 °C at 12,000 r/min, freeze-dried for 12 h. The collected product was the SiO_2_@CMCS-*N*-2-HACC nanoparticles loaded with inactivated NDV (NDV/SiO_2_@CMCS-*N*-2-HACC). The preparation process is shown in Figure 1.

### 2.6. Analysis of Physical and Chemical Properties of the SiO_2_@CMCS-N-2-HACC Nanoparticles

The transmission electron microscope (Hitachi, Tokyo, Japan) and scanning electron microscope image of the sample was recorded (voltage 200 kV) (Thermo Fisher Scientific, New York, NY, USA). Dynamic light scattering (DLS model is BI-Zeta PALS, which is produced by Brookheven Instruments, New York, NY, USA) was used to measure the particle size of nanoparticles in neutral and low-concentration solution. Zeta potential was used to measure the charge number of nanoparticles through dynamic light scattering. A small angle X-ray powder diffraction (XRD) pattern was obtained on the D/Max IIIA (Rigaku) diffractometer (Bruker, Leipzig, Germany). The formula was as follows:2dsinθ = nλ(3)
where θ is the incident angle, d is the interplanar spacing, n is the diffraction order, and λ is the wavelength of the incident ray.

Brunauer-Emmett-Teller (BET) method was used to analyze the pore volume and pore size. Place approximately 0.17 g of the sample in a clean sample tube and treat it at 250 °C for 2 h. After accurate weighing, the N_2_ adsorption-desorption test was carried out at the temperature of liquid nitrogen. After the test, we used the BELmaster software that came with the instrument to analyze. The pore size of the sample could be obtained according to the BET algorithm. The formula was as follows:(4)$$\mathrm{BET}\;\mathrm{equation}:\;\frac{p}{v\left(p_{0}-p\right)}=\frac{1}{v_{m}c}+\frac{c-1}{v_{m}c}\left(\frac{p}{p_{0}}\right)$$
(5)$$\mathrm{BJH}\;\mathrm{equation}:\;V_{pn}=R_{n}\Delta V_{n}-R_{n}\Delta t_{n}c\sum_{j=1}^{n-1}A_{PJ}$$
(6)$$\mathrm{Kelvin}\;\mathrm{equation}:\;ln\frac{p}{p_{0}}=-\frac{2\gamma V_{L}}{RT}\frac{1}{r_{m}}$$
where *p* is the equilibrium pressure of the adsorbed gas at the adsorption temperature; *p*_0_ is the saturated vapor pressure; *v* is the total volume of the adsorbed gas at the equilibrium pressure; *v_m_* is the volume of the gas covered on the solid surface; *c* is the adsorption-related constant; *γ* is the surface tension; *R* is Gas constant; *V_L_* is the molar volume of the adsorbate liquid phase; *r_m_* is the pore size.

### 2.7. Cytotoxicity of the NDV/SiO_2_@CMCS-N-2-HACC

CCK-8 Kit was used to determine the cytotoxicity of the NDV/SiO_2_@CMCS-*N*-2-HACC to DF-1 cells (Beyotime, Shanghai, China). OD_450_ was measured with a microplate reader, and cell viability (%) calculated with the following formula:Cell viability (%) = [A_S_ − A_b_]/[A_C_ − A_b_] × 100%(7)
where As is the absorbance of the well with cells, CCK-8 solution and the sample solution to be tested; A_b_ is the absorbance of the well with the culture medium and CCK-8 solution but no cells; Ac is the absorbance of the well with cells, CCK-8 solution and no test the absorbance of the well of the sample solution.

### 2.8. Integrity Assay of NDV Encapsulated in the SiO_2_@CMCS-N-2-HACC Nanoparticles

After testing the toxicity of the nanoparticles, the integrity of NDV in the nanoparticles should also be tested to ensure that NDV was not destroyed in the nanoparticles. SDS-PAGE was used to determine the integrity of the Newcastle disease inactivated vaccine in the composite nano-vaccine (Thermo Fisher Scientific, New York, NY, USA). Coomassie brilliant blue staining (0.25%) was selected for protein staining. It was then decolored with an aqueous solution containing methanol (40%) and acetic acid (10%). The observation time was 1 h.

### 2.9. Stability of the SiO_2_@CMCS-N-2-HACC Nanoparticles and SiO_2_ Nanoparticles Coated Antigen

#### 2.9.1. Determination of Acid Resistance

A total of 500 mg of SiO_2_@CMCS-*N*-2-HACC nanoparticles and SiO_2_ nanoparticles with NDV were added to 50 mL of artificial gastric juice (pH = 1.2), and then shaken in a water bath shaker at 37 °C at 180 r/min. Samples were taken at 0, 1, 2, 3, and 4 h to determine the light transmittance and the acid resistance of NDV/SiO_2_@CMCS-*N*-2-HACC analyzed. At each time point it was repeated five times in parallel.

#### 2.9.2. Determination of Enteric Solubility

In total, 1.5 grams of SiO_2_@CMCS-*N*-2-HACC nanoparticles and SiO_2_ nanoparticles containing NDV were added to 50 mL of artificial intestinal fluid (pH = 6.8), and shaken in a water bath shaker at 37 °C at 180 r/min. Samples were taken at 0, 0.5, 1, 2, 4, 8, 12, 24, 36, 48, 72 h to determine the light transmittance, and the dissolution of NDV in SiO_2_@CMCS-*N*-2-HACC nanoparticles and SiO_2_ nanoparticles analyzed. At each time point it was repeated five times in parallel.

#### 2.9.3. Determination of Bile Salt Tolerance

A total of 500 mg of SiO_2_@CMCS-*N*-2-HACC nanoparticles and SiO_2_ nanoparticles containing NDV were added to 50 mL of artificial bile salt solution, and shaken in a water bath shaker at 37 °C at 180 r/min. Samples were taken at 0, 1, 2, 3, and 4 h to determine the light transmittance of the samples, and the bile salt tolerance of NDV in SiO_2_@CMCS-*N*-2-HACC nanoparticles and SiO_2_ nanoparticles analyzed. At each time point it was repeated five times in parallel.

#### 2.9.4. Influence of pH and Ion Concentration on Stability

A total of 100 mg of NDV coated SiO_2_@CMCS-*N*-2-HACC nanoparticles and NDV coated SiO_2_ nanoparticles were dispersed into HCl solution with pH values of 1, 2, 3, 4, 5, 6, and 7, respectively. A further 100 mg of SiO_2_@CMCS-*N*-2-HACC nanoparticles and SiO_2_ nanoparticles containing NDV was dispersed in Tris-HCl buffer (pH = 7.4), NaCl solution with different ion concentrations (0, 0.2, 0.4, 0.6, 0.8, 1.0 mol/L). After standing for 2 days at 4 °C at 12,000 r/min for 30 min, they were freeze dried for 12 h. The product was collected, weighed, and the remaining nanoparticles were calculated as a percentage of the original mass. It was repeated for five times in parallel at each time point.

### 2.10. In Vitro Release

A total of 10 mg of NDV/SiO_2_@CMCS-*N*-2-HACC nanoparticles and NDV/SiO_2_ nanoparticles were weighed and added to 2 mL phosphate buffer (pH = 7.4, pH = 8.6) and acetate buffer (pH = 3, pH = 5) respectively, and placed at 37 °C at 100 r/min in a constant temperature oscillator at different times (0, 3, 6, 9, 12, 24, 36, 48, 60 h). 1 mL supernatant was taken out (while adding 1mL buffer of the same pH) and centrifuged at 12,000 r/min for 30 min at 37 °C. Then, 10 microliters of the supernatant were taken and the bovine serum albumin kit (Thermo Fisher Scientific, New York, NY, USA) was used to detect the NDV content in SiO_2_@CMCS-*N*-2-HACC nanoparticles, and each sample was tested three times in parallel. In vitro release characteristic curve was plotted as the percentage of NDV cumulative release in the total amount versus time.

## 3. Results

### 3.1. Characterization of the Mesoporous SiO_2_ Nanoparticles

The mesoporous SiO_2_ nanoparticles synthesized under the optimal process conditions were spherical in shape and uniform in size (Figure 2A,B). The mesopores on the surface of the SiO_2_ nanoparticles were clearly visible. SEM and TEM confirmed the presence of mesopores on the surface of SiO_2_ nanoparticles.

It can be seen from Figure 2C that when the 2θ was 0.3883°, the XRD pattern had an obvious peak, which proved that the synthesized SiO_2_ nanoparticles had mesopores. The mesopore size of mesoporous SiO_2_ nanoparticles was calculated according to the Bragg equation, where λ = 1.54 nm and 2θ = 0.3883. The calculated mesopore was 4.0 nm.

The isotherm of the synthesized mesoporous SiO_2_ nanoparticles was IV-type isotherm, and a retention ring appeared in the middle of the isotherm (Figure 2D). This is because the mesoporous SiO_2_ nanoparticles undergo capillary condensation and the isotherm rise rapidly. According to the capillary theory, the hysteresis loop will only appear when the desorption pressure of the same adsorption amount is lower than the adsorption pressure. When the relative pressure P/P_0_ was over 0.6, the mesoporous SiO_2_ nanoparticles underwent capillary condensation, so the adsorption line and the desorption line did not overlap, and the desorption isotherm was above the adsorption isotherm.

It can be seen from Figure 2E that there was a vertex at the position of about 4 nm on the abscissa, which proved that the mesoporous SiO_2_ nanoparticles had mesopores with a pore diameter of 3.7 nm. These results were consistent with the XRD small-angle diffraction. The specific surface area of the sample was 367 m^2^/g.

### 3.2. Characterization of the Aminated Mesoporous SiO_2_ Nanoparticles

The strong and broad absorption peak at 1090 cm^−1^ was the antisymmetric stretching vibration peak of the Si–O–Si group, and the peak at 807 cm^−1^ was the symmetric stretching vibration peak of the Si–O bond (Figure 2F-b). The peak near 1638 cm^−1^ belonged to the H–O–H bending vibration peak of water, and the O-H stretch vibration peak at 3400 cm^−1^. The peak at 697 cm^−1^ in Figure 2F-a was the N-H bending vibration peak, and the peak at 1500 cm^−1^ was the symmetrical vibration peak of -NH_3_^+^. These results indicated that the mesoporous SiO_2_ nanoparticles have been aminated.

As shown in Figure 3, the unaminated mesoporous SiO_2_ nanoparticles were intact (Figure 3A,C), uniformly distributed, and had a moderate particle size, with a particle size of 213.4 ± 6.5 nm; The porous SiO_2_ nanoparticles were uniformly distributed, and were spherical in shape (Figure 3B,D) and moderate in particle size, with a particle size of 208.0 ± 20.5 nm. The unmodified mesoporous SiO_2_ was negatively charged [44,45,46], but as a mucosal vaccine carrier, only positively charged could it have mucosal adhesion. So that the SiO_2_ nanoparticles were aminated. It can be seen from Figure 3E that the aminated mesoporous SiO_2_ nanoparticles showed a positive charge with a potential of 17.4 ± 1.7 mV.

### 3.3. Characterization of the NDV/SiO_2_ Nanoparticles

As shown in Figure 4A, there were some small particles on the surface of the NDV/SiO_2_ nanoparticles. The small particles were NDV, indicating that NDV and aminated mesoporous SiO_2_ nanoparticles were bonded together by electrostatically adsorption. NDV and aminated mesoporous SiO_2_ nanoparticles had good electrostatic adsorption effect.

### 3.4. Physiochemical Characteristics of the NDV/SiO_2_@CMCS-N-2-HACC

As shown in Figure 4B, the NDV/SiO_2_@CMCS-*N*-2-HACC had a complete spherical shape, uniform distribution of nanoparticles, an average particle size of 243.0 ± 27.0 nm, and no adhesion between the nanoparticles; *N*-2-HACC and CMCS were successfully cross-linked and coated on the surface of NDV/SiO_2_ nanoparticles. The surface of NDV/SiO_2_@CMCS-*N*-2-HACC was smooth and no obvious mesopores were seen (Figure 4C), indicating that *N*-2-HACC and CMCS had been successfully cross-linked, which was mutually confirmed by observation. Transmission electron microscope results further proved that *N*-2-HACC and CMCS were cross-linked and successfully coated on the surface of NDV/SiO_2_ nanoparticles. As shown in Figure 4D, the NDV/SiO_2_@CMCS-*N*-2-HACC had positive electrical property and the Zeta potential was 12.1 ± 1.3 mV.

When the mass of the aminated mesoporous SiO_2_ nanoparticles were 5 mg, the encapsulation efficiency and loading capacity of the nanoparticles were the highest (Figure 5). When the mass of aminated mesoporous SiO_2_ nanoparticles was less than 5 mg, the amount of NDV inactivation solution added was more than the mass of aminated mesoporous SiO_2_ nanoparticles, which led to the release of part of NDV. The encapsulation efficiency was the smallest when the aminated mesoporous SiO_2_ nanoparticles were 7 mg, which might be because the mass of the aminated mesoporous SiO_2_ nanoparticles was larger than the amount of the NDV inactivation solution. Therefore, the amount of NDV supported by each aminated mesoporous SiO_2_ nanoparticles was reduced, resulting in a decrease the weight of aminated mesoporous SiO_2_ nanoparticles, and the low-weight aminated mesoporous SiO_2_ nanoparticles was not separated during centrifugation, so the encapsulation efficiency and loading capacity were reduced. Therefore, we determined that 1 mL of NDV inactivation solution (virus content of 7.504 mg/mL) was added with 5 mg of aminated mesoporous SiO_2_ nanoparticles.

### 3.5. Cytotoxicity of the NDV/SiO_2_@CMCS-N-2-HACC

In order to verify whether the synthesized nanomaterials were irritating to animals, the cytotoxicity of the nanomaterials was tested in this experiment. Figure 6 showed that the survival rate of DF-1 cells was very high under the different concentrations of SiO_2_@CMCS-*N*-2-HACC nanoparticles, and the overall survival rate was as high as 85.3 ± 6.3 %; When the concentration reached 1000 μg/mL, the survival rate of DF-1 cells was still very high, the cell survival rate was 89.1 ± 2.4 %, indicating that the nano delivery carrier constructed in this study had very low toxicity or was non-toxic, safe, and could be used for drug delivery [46].

### 3.6. Integrity of NDV in the SiO_2_@CMCS-N-2-HACC

NDV has six structural proteins, namely high molecular weight protein (L), hemagglutinin-neuraminidase protein (HN), fusion protein (F), nucleoprotein (NP), phosphoprotein (P) and matrix protein (M). The relative molecular weight of L protein is about 150 KD, with light coloration; HN protein has a molecular weight of 75 kD; F protein has a molecular weight of 54 kD; NP protein has a molecular weight close to that of P protein, 47 kD and 45 kD, respectively. They are clustered together during electrophoresis and are difficult to separate; The M molecular weight of the protein is 41 kD. Figure 7 showed that the antigen strips of bands 1 and band 2 were the same. Four bands were seen at 75, 54, 45, 41 kD. It showed that SiO_2_@CMCS-*N*-2-HACC could protect the integrity of the antigen from being destroyed, and the biological activity of the antigen was well maintained.

### 3.7. Stability of the SiO_2_@CMCS-N-2-HACC Nanoparticles and SiO_2_ Nanoparticles Coated Antigen

As shown in Figure 8A, the light transmittance of the two materials in artificial gastric juice remained above 84% after 4 h, indicating that NDV/SiO_2_@CMCS-*N*-2-HACC and NDV/SiO_2_ nanoparticles were not abundant in gastric juice break down. They could resist the acidic environment in the stomach and may enter the small intestine through the stomach environment. SiO_2_@CMCS-*N*-2-HACC nanoparticles had better gastric acid resistance than SiO_2_ nanoparticles and could better protect the antigen. So that the antigen was not broken down.

Figure 8B showed that as the treatment time increases, the light transmittance of NDV/SiO_2_@CMCS-*N*-2-HACC and NDV/SiO_2_ nanoparticles in artificial intestinal juice gradually decreases. However, the light transmittance of NDV/SiO_2_@CMCS-*N*-2-HACC dropped sharply within 2 h, possibly because the CMCS and *N*-2-HACC coated with NDV/SiO_2_ nanoparticles disintegrated, which released a large amount of NDV. The antigen can quickly act on the small intestine. After the carrier was used in vaccine preparation, it may have a rapid and long-term effect on the body. However, NDV/SiO_2_ nanoparticles had no effect of sudden release, and the release amount was almost the same as NDV/SiO_2_@CMCS-*N*-2-HACC. It showed that SiO_2_@CMCS-*N*-2-HACC nanoparticles can better act on the small intestine. May have the potential to produce a targeted effect in the small intestine.

In order to test the stability of NDV/SiO_2_@CMCS-*N*-2-HACC and NDV/SiO_2_ nanoparticles in bile salts, this experiment measured the bile tolerance of the two materials. Figure 8C showed that when the two kinds of nanoparticles were treated in artificial bile for 4 h, the light transmittance was above 85%, but SiO_2_@CMCS-*N*-2-HACC nanoparticles had better tolerance in bile than SiO_2_ nanoparticles. The tolerance of the vaccine carrier can avoid the corrosion of bile salts, reach the intestinal tract smoothly, and well maintain the biological activity of the vaccine.

### 3.8. Influence of pH and Ion Concentration on the Stability of SiO_2_@CMCS-N-2-HACC Nanoparticles and SiO_2_ Nanoparticles Coated Antigen

To test the influence of pH on the material, the original mass was set as V in this experiment, the mass after 48 h of treatment was V_0_, and the graph was drawn with (V_0_/V) × 100% as the ordinate and time as the abscissa. It can be seen from Figure 8D that the products were collected after being placed in the HCl solution for 48 h. The product mass was more than 36% of the original mass. As the pH in the HCl solution increased, the remaining NDV/SiO_2_@CMCS-*N*-2-HACC and NDV/SiO_2_ nanoparticles masses gradually increased. However, when the pH was lower, NDV/SiO_2_@CMCS-*N*-2-HACC had more residual mass than NDV/SiO_2_ nanoparticles, indicating that SiO_2_@CMCS-*N*-2-HACC nanoparticles has better tolerance in acidic environment. When the pH was 7.0 and 7.4, both materials had almost no mass loss, which indicates that SiO_2_@CMCS-*N*-2-HACC nanoparticles and SiO_2_ nanoparticles cannot exist for a long time in acidic solutions, but they can exist for a long time under neutral and weak alkaline conditions.

In order to explore the effect of ion concentration on materials, set the original mass as M, and the mass after 48 h of processing as M_0_. The graph uses (M_0_/M) × 100% as the ordinate and ion concentration as the abscissa. It can be seen from Figure 8E that the weight of NDV/SiO_2_@CMCS-*N*-2-HACC and NDV/SiO_2_ nanoparticles were almost unchanged after being placed in different concentrations of NaCl solution for a period of time. The results showed that the ion concentration had little effect on the stability of SiO_2_@CMCS-*N*-2-HACC nanoparticles and SiO_2_ nanoparticles. Both materials can be stored in NaCl solution.

### 3.9. In Vitro Release of the SiO_2_@CMCS-N-2-HACC Nanoparticles and SiO_2_ Nanoparticles Coated Antigen

Figure 8F shows the effect of different pH on the vitro release of NDV/SiO_2_@CMCS-*N*-2-HACC. The release rate of nanoparticles was the fastest in the first 10 h, and gradually slowed down after 2 h, then the release rate reached the limit after the 30 h of the reaction. When the pH was neutral (pH = 7.4) and weakly acidic (pH = 5), the cumulative release of nanoparticles reached 91.3% and 83.1%. However, when the pH was strong acid (pH = 3) and alkaline (pH = 8.6), the cumulative release of nanoparticles will be reduced to 59.4% and 40.2%. Figure 8G shows the effect of different pH on the vitro release of NDV/SiO_2_ nanoparticles. When the pH value was 7.4, the cumulative release reaches 91.1%, when the pH was 3 and 5, the cumulative release was above 60%, and when the pH was 8.6, the cumulative release was the lowest. It can be seen from Figure 8F,G that both types of nanoparticles had properties of sustained-release and pH-dependency. They released more NDV in neutral and weakly acidic environments, but less in strong acid and alkaline environments. The total release rate of NDV/SiO_2_@CMCS-*N*-2-HACC under different pH were higher than that of NDV/SiO_2_ nanoparticles. The experimental results were consistent with those of nanoparticle tolerance test.

## 4. Conclusions

In many previous reports, it has been found that mesoporous SiO_2_ nanoparticles and aminated mesoporous SiO_2_ nanoparticles are better delivery vehicles. Nanoparticles synthesized by CMCS and *N*-2-HACC also have the potential of delivery carriers. In this study, the aminated mesoporous SiO_2_ nanoparticles and CMCS-*N*-2-HACC nanoparticles were combined to prepare the SiO_2_@CMCS-*N*-2-HACC nanoparticles. The SiO_2_@CMCS-*N*-2-HACC nanoparticles has better physical and chemical characteristics than aminated mesoporous SiO_2_ nanoparticles. The results in this study showed that the encapsulation efficiency and drug loading of the SiO_2_@CMCS-*N*-2-HACC nanoparticles were relatively high, non-toxic, with sustained release effect and mucosal adhesion. It has great potential as vaccine and/or drug delivery.

## Figures and Tables

**Figure 1 molecules-26-02490-f001:**
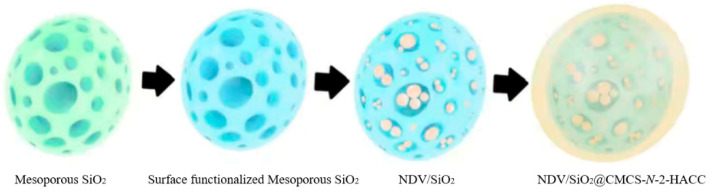
3D model diagram from mesoporous SiO_2_ nanoparticles to NDV/SiO_2_@CMCS-*N*-2-HACC.

**Figure 2 molecules-26-02490-f002:**
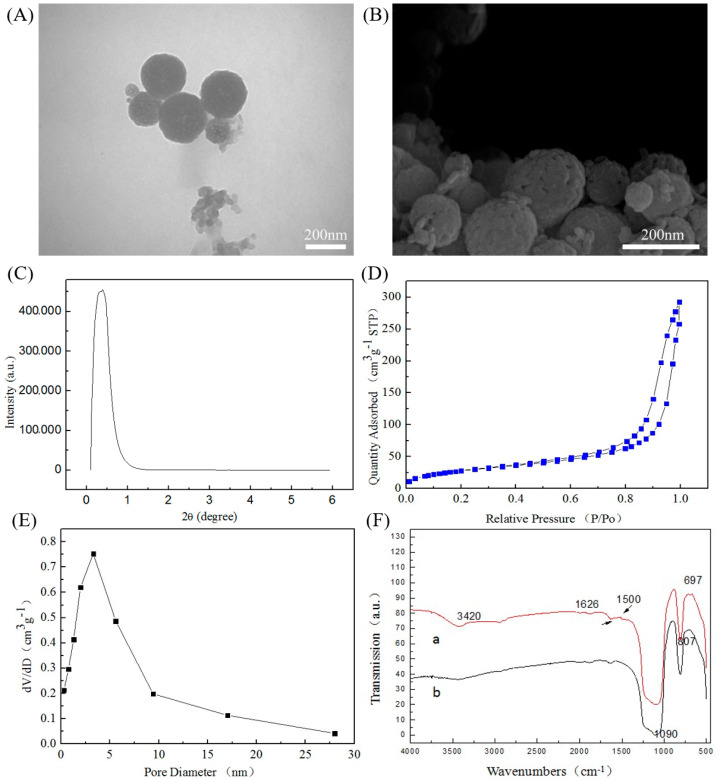
The characteristics of mesoporous SiO_2_ nanoparticles: (**A**) transmission electron microscope; (**B**) scanning electron microscope; (**C**) XRD pattern of mesoporous SiO_2_ nanoparticles; (**D**) mesoporous SiO_2_ nanoparticle isotherm; (**E**) BET pore size analysis chart of mesoporous SiO_2_ nanoparticles; (**F**) infrared spectrum of aminated mesoporous SiO_2_ nanoparticles: (**a**) mesoporous SiO_2_ nanoparticles after amination; (**b**) mesoporous SiO_2_ nanoparticles without amination.

**Figure 3 molecules-26-02490-f003:**
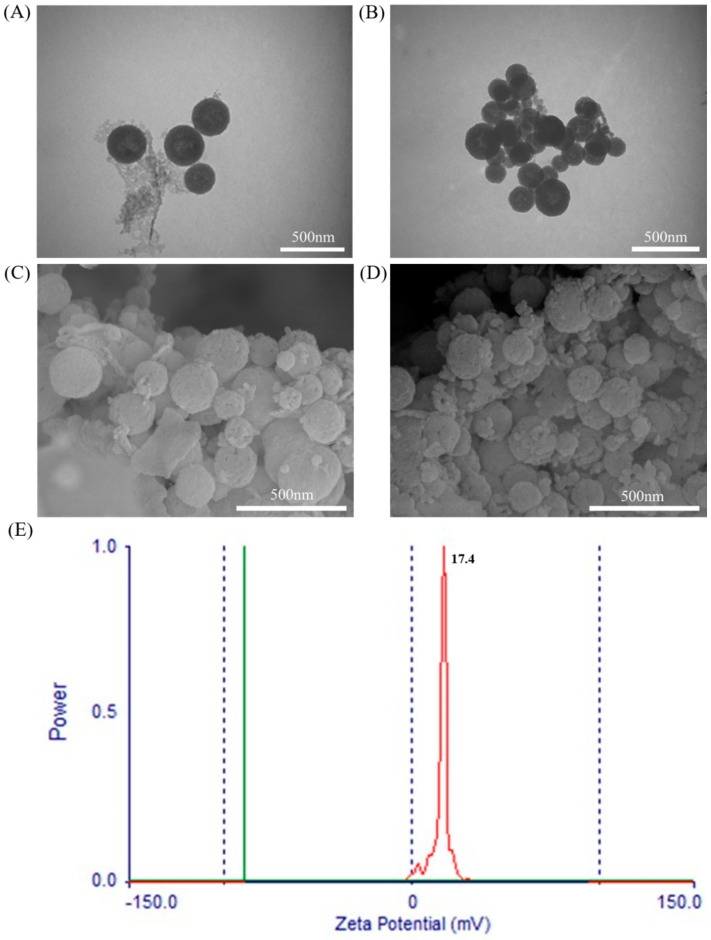
Transmission electron micrograph (TEM) and scanning electron micrograph (SEM) of aminated mesoporous SiO_2_ nanoparticles: (**A**) TEM image of the unaminated mesoporous SiO_2_ nanoparticles; (**B**) TEM image of mesoporous SiO_2_ nanoparticles after amination; (**C**) SEM image of the unaminated mesoporous SiO_2_ nanoparticles; (**D**) after amination SEM image of mesoporous SiO_2_ nanoparticles; (**E**) Zeta potential of aminated mesoporous SiO_2_ nanoparticles.

**Figure 4 molecules-26-02490-f004:**
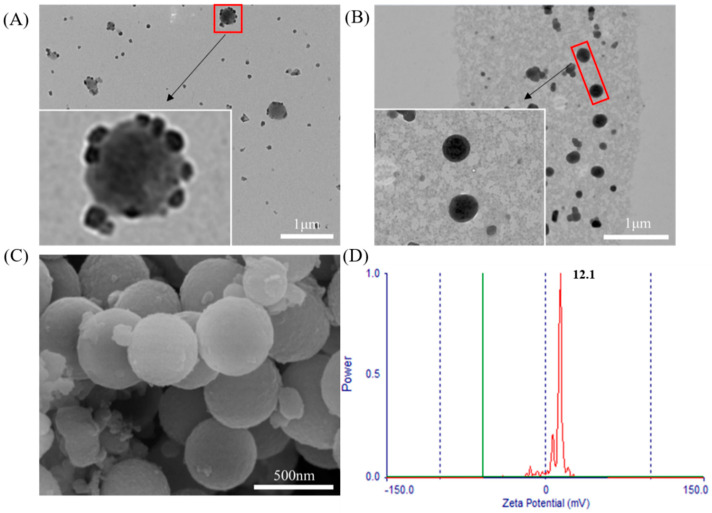
Transmission electron microscope morphology of NDV/SiO_2_ nanoparticles: (**A**) transmission electron microscope image of NDV/SiO_2_ nanoparticles; (**B**) NDV/SiO_2_@CMCS-*N*-2-HACC transmission electron microscope; (**C**) scanning electron micrograph of NDV/SiO_2_@CMCS-*N*-2-HACC; (**D**) Zeta potential diagram of NDV/SiO_2_@CMCS-*N*-2-HACC.

**Figure 5 molecules-26-02490-f005:**
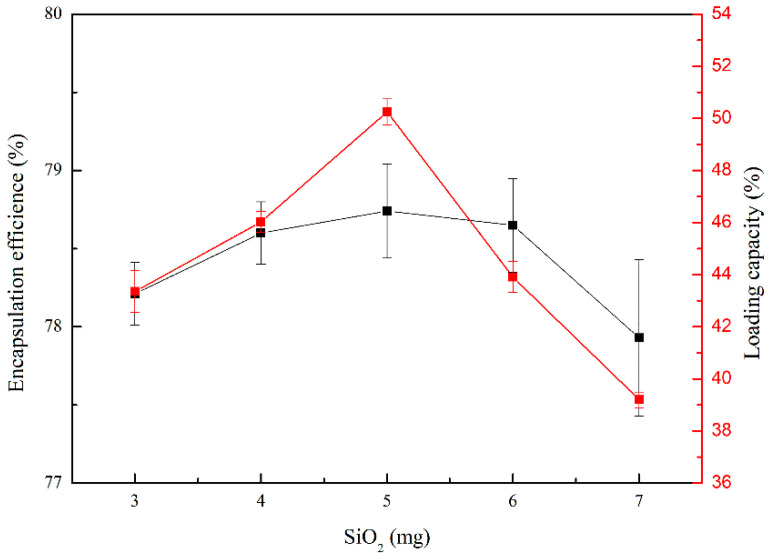
Determination of the encapsulation efficiency and loading capacity of aminated mesoporous SiO_2_ nanoparticles adsorbed by Newcastle disease virus inactivation solution.

**Figure 6 molecules-26-02490-f006:**
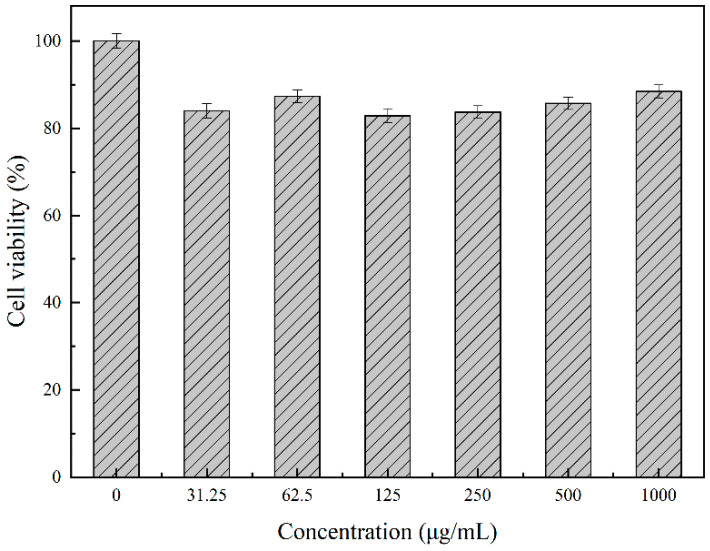
Detection of DF-1 cytotoxicity at different concentrations of NDV/SiO_2_@CMCS-*N*-2-HACC.

**Figure 7 molecules-26-02490-f007:**
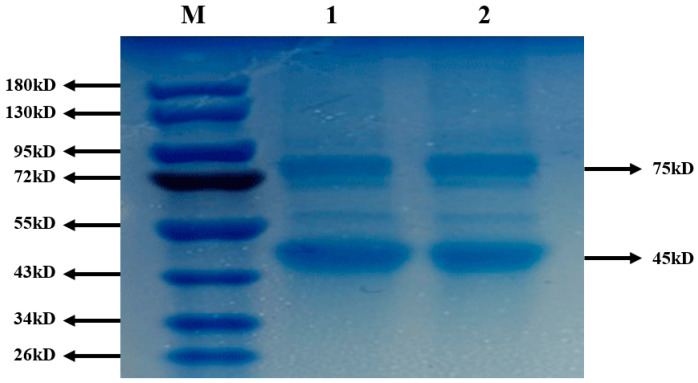
Detection of integrity of NDV in SiO_2_@CMCS-*N*-2-HACC by Western blotting: (**M**) protein marker; (**1**) inactivated NDV; (**2**) recovery in NDV/SiO_2_@CMCS-*N*-2-HACC.

**Figure 8 molecules-26-02490-f008:**
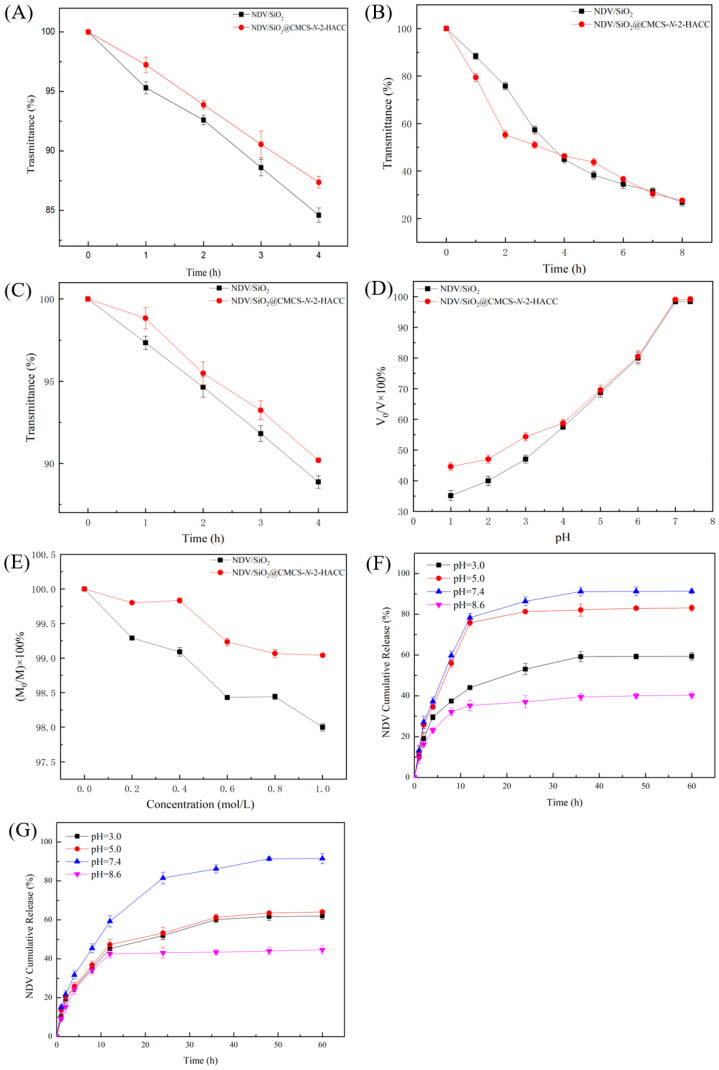
Stability of SiO_2_@CMCS-*N*-2-HACC nanoparticles and SiO_2_ nanoparticles coated antigen: (**A**) acid resistance test of NDV/SiO_2_@CMCS-*N*-2-HACC and NDV/SiO_2_ nanoparticles; (**B**) determination of enteric solubility of NDV/SiO_2_@CMCS-*N*-2-HACC and NDV/SiO_2_ nanoparticles, (**C**) NDV/SiO_2_@CMCS-*N*-2-HACC and NDV/SiO_2_ nanoparticles bile salt resistance test; (**D**) effect of different pH on the stability of NDV/SiO_2_@CMCS-*N*-2-HACC and NDV/SiO_2_ nanoparticles; (**E**) effect of different ionic strengths on the stability of NDV/SiO_2_@CMCS-*N*-2-HACC nanoparticles and NDV/SiO_2_ nanoparticles; (**F**) in vitro release of NDV/SiO_2_@CMCS-*N*-2-HACC under different pH; (**G**) in vitro release of NDV/SiO_2_ nanoparticles under different pH.

## Data Availability

The data presented in this study are availability within the article.

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
