# Peer review of "Evaluation of Chitosan Derivatives Modified Mesoporous Silica Nanoparticles as Delivery Carrier"

_molecules, 2021, doi:10.3390/molecules26092490_

Round 1
Reviewer 1 Report
The authors have successfully addressed all my concerns, improving the manuscript with their edits. In my opinion, the manuscript is now acceptable for publication.
Reviewer 2 Report
To the authors,
Thank you for resubmitting the revised version of this manuscript. The provided revisions have enhanced the quality of the manuscript.
Regards
This manuscript is a resubmission of an earlier submission. The following is a list of the peer review reports and author responses from that submission.
Round 1
Reviewer 1 Report
In this manuscript, Li et al. developed chitosan derivatives modified mesoporous silica nanoparticles as delivery carrier. In general, the experiments in the manuscript were well-designed and the data was clearly presented. The authors also provided detailed experimental procedures in the experimental sections so that other researchers could easily repeat the experiments. In addition, the data can also support the conclusion. Therefore, I recommend the paper to be published in Molecules after the minor revision.
1. In line 239 and 243, it should be capillary condensation instead of capillary agglomeration. The author should revise it accordingly.
2. In Figure 2e, the author should state the pore size distribution was calculated by BJH or DFT method. BET is not the method to calculate the pore size distribution. Instead, it is a method to calculate the surface area of the nanomaterials. If possible, please also show the pore volume of the materials so that other readers could evaluate the efficacy for drug loading.
3. In line 277, the author should provide the zeta potential value of unmodified mesoporous SiO2 nanoparticles and compared the value with the aminated mesoporous SiO2 nanoparticles. The following references should also be cited to validate the correctness of the zeta potential values. (J. Am. Chem. Soc. 2019, 141, 17670-17684; ACS Nano, 2019, 13, 1292-1308)
4. In Figure 5, the author showed the encapsulation rate as a function of the mass of nanoparticles used. Are there any significant differences of encapsulation rate between these different conditions?
5. In Figure 2a, can the author provide a TEM image which can show the clear porous structure?
6. In Figure 6, can the author explain why the viability of the cells was higher than 100% when there was no treatment (0 ug/mL). It is usually 100% and serves as a control. (ACS Appl. Mater. Interfaces, 2017, 9, 22235-22251)
7. Several papers related to the biomedical applications of mesoporous nanoparticles should also be cited in the references:
ACS Nano 2013, 7, 8423-8440.
J. Am. Chem. Soc.2014, 136, 16326–16334.
Nat. Commun. 2017, 8, 357.
Acc. Chem. Res. 2019, 52, 1531-1542.
ACS Appl Mater Interfaces, 2018, 10, 31870-31881.
Reviewer 2 Report
This manuscript is focused on the preparation and characterization of mesoporous SiO2 covered with two chitosan derivatives (N-2-Hydroxypropyl trimethyl ammonium chitosan chloride and carboxymethyl chitosan) as a career for Newcastle disease virus (NDV). In general, the paper is written and prepared carelessly and therefore it is vague. The paper could have some scientific contribution, but it has to be significantly improved or rewritten. The material part must be described and presented in a reliable and accurate manner allowing others to repeat the presented research.
Specific comments:
- First of all, the paper must be edited and proofread for appropriate use of the English language. There are too many English errors and wrong constructions, which sometimes drastically affect the scientific meaning, making the paper difficult to read and understand. A native English speaker with a scientific background should carefully revise the manuscript prior to its resubmission.
- The structure of the article is not optimal – it consists of a large number of micro-sections poorly linked to each other. The paper structure must be reconsidered.
- In the Abstract (lines 14-16), the authors stated that the chitosan derivatives had been synthesized in their lab. However, in the Materials section, they mentioned that chitosan derivatives were purchased from commercial sources (lines 89-92). Where is the correct statement?
- Lines 34-35: What do you mean by “The drug delivery system … regulates the distribution of organisms in the body”?
- Lines 47-48: I'm afraid I have to disagree with your statement that mesoporous SiO2 “has been proven to be an ideal drug delivery system”. In the introduction, it is necessary to consider both the advantages and disadvantages of mesoporous SiO2 as a drug delivery system. In my opinion, there are some issues with biodegradation, accumulation in organs (e.g., in the liver), and excretion of mesoporous SiO2.
- You cannot just say "simple and convenient advantages", you have to explain what these advantages are.
- Section 2.1. Materials: The chitosan derivatives must be thoroughly characterized regarding their molecular weight (by viscometry, light scattering, or size exclusion chromatography), the degree of substitution, and the degree of deacetylation (by NMR, IR, elemental analysis, or titration). Also, indicate the source of chitosan (crab, shrimp, fungi, etc.). The properties of chitosan and its derivatives are very dependent on these parameters; therefore, the European Chitin Society does not recommend publication of the research papers on chitosan, in which this information is missing.
- Section 2.3.2. Verification of amino chain of mesoporous SiO2 nanoparticles: The description of the method does not include the verification method itself.
- Section 2.6. Analysis of physical and chemical properties of the SiO2@CMCS-N-2-HACC: The description of the equipment and conditions for DLS experiments and zeta potential measurements is missing. There is no description of the sorption experiment (BET).
- Sections 2.10 and 3.9. In vitro release: The choice of NDV release kinetics in PBS with pH=7.4 has not been justified by the authors. Since oral administration is intended, there must be a different protocol.
- The authors repeatedly state that specific experiments were conducted under "optimal conditions" (lines 113, 171, 255, 288, or " best ratio" (line 280); however, they do not provide the readers with a selection of these optimal conditions.
- Across the text: The standard deviation should be expressed as one significant figure; that is, unless the number is between 11 and 19 times some power of ten, in which case you can use two significant figures. The mean value should be rounded off at the decimal place corresponding to the last significant digit of its standard deviation.
Reviewer 3 Report
To the authors,
The current manuscript is describing a novel approach for designing a nanoplatform for an orally administered vaccine. The authors could successfully hypothesize the ideas and answer to the main questions, however, there are some comments to enhance the quality of the paper:
1- The quality of figures are not good. And should be improved.
2- the authors should explain how they used XRD to determine the pore size because this is not a common method.
3- in Fig 3 and 4, the value of the ZP should be determined on the graph for the ease of understanding.
4- Fig 5, is this an encapsulation rate or percentage? If it is a rate, it is compared to what value?
5- Why the authors used DF-1 cells? It was preferred to see the results on human cell line.
Regards
Reviewer 4 Report
The authors present interesting sets of data on the preparation of chitosan derivates modified mesoporous SiO2 as an oral carrier of Newcastle disease virus. The drawback of the paper is that the authors did not evaluate the efficacy of the nanoparticles. However, the experimental data are comprehensive, and the experiments were appropriately performed. For this reason, the submitted paper is worthy of publication. Prior to publication the authors are requested to address the following issues.
- Overall English should be improved.
- The use of silica, SiO2, SiO2 nanomaterials, and SiO2 nanoparticles should be consistent throughout the manuscript.
- (line 13) An antimicrobial property may not be mentioned as this property was not related to the study.
- (lines 39-42) These statements are not convincing and should be stated in relation to scientific motivation of the current study.
- (lines 47-48) Mesoporous SiO2 may not be regarded as “a drug delivery system” (“A drug delivery agent” is fine as mentioned in line 61).
- (line 64) “better performance” should be specifically stated.
- (lines 82-83) How uniform, how large surface area, and how large pore volume of SiO2 are suitable or required? Please compare these properties in comparison to other mesoporous particles. Please also provide the scientific background of the selection of SiO2 among other mesoporous particles.
- (line 128) What were the particle sizes of Newcastle disease virus?
- (line 134) In addition to the encapsulation efficiency, encapsulation amount in SiO2 nanoparticles should be evaluated. Please also provide suitable equation to calculate encapsulation efficiency and amount.
- (line 157) In addition to particle sizes of nanoparticles, the pore sizes that accommodate viruses should be measured.
Round 2
Reviewer 2 Report
Several significant problems have not been addressed in the revised manuscript; therefore, I can not recommend the manuscript for publication in Molecules:
Major issues:
- The chitosan sample must be carefully characterized. The properties of chitosan are very dependent on these parameters; therefore, the wide ranges of values provided by the manufacturer (like degree of deacetylation 80%-95%, the viscosity is 50-800mPa·s; line 145) are clearly insufficient. The molecular weight of chitosan indicated by the authors as 1526.45 is quite surprising both in magnitude and accuracy of its measurement. Have you worked with chitooligosaccharides (average degree of polymerization 9-10)? What method of molecular weight determination did you use?
- Lines 149-151: The chitosan derivatives (N-2-HACC and CMCS) are also poorly characterized. The molecular weight of the samples and the degree of deacetylation are not given. Characterization of the polymer samples you are working with is crucial because the reproducibility of your results essentially depends on the polymer characteristics.
- I am not satisfied with your answer to question 10. Sections 2.10 and 3.9. In vitro release: The choice of NDV release kinetics in PBS with pH=7.4 has not been justified by the authors. Since oral administration is intended, there must be a different protocol.
The study of the stability of the SiO2@CMCS-N-2-HACC system under different conditions is in no way related to the study of NDV release kinetics under oral administration conditions. Again, the choice of PBS with pH=7.4 is not justified by the authors.
Minor issues:
- I would suggest removing these headings:
3.4.1 Morphology and particle size of the NDV/SiO2@CMCS-N-2-HACC (line 454)
3.4.2 Determination of the ratio of aminated mesoporous SiO2 nanoparticles to NDV (line 465)
- Line 178: What “amino chain” do you mean? I do not expect any amino chain formation upon reaction of silica with γ-aminopropyltriethoxysilane.